# Gapful electrons in a vortex core in granular superconductors

Dmitry E. Kiselov[1], Mikhail A. Skvortsov[1] and Mikhail V. Feigel'man[1,2]

**1** L. D. Landau Institute for Theoretical Physics, 142432 Chernogolovka, Russia
**2** Floralis & LPMMC, Universite Grenoble - Alpes, France

## Abstract

We calculate the quasiparticle density of states (DoS) inside the vortex core in a granular superconductor, generalizing the classical solution applicable for dirty superconductors. A discrete version of the Usadel equation for a vortex is derived and solved numerically for a broad range of parameters. Electron DoS is found to be gapful when the coherence length $\xi$ becomes comparable to the distance between neighboring grains $l$. Minigap magnitude $E_g$ grows from zero at $\xi \approx 1.4l$ to third of superconducting gap $\Delta_0$ at $\xi \approx 0.5l$. The absence of low-energy excitations is the main ingredient needed to understand strong suppression of microwave dissipation recently observed in a mixed state of granular Al.



## 1 Introduction

Electron-hole quasiparticle states inside the core of an Abrikosov vortex in a clean superconductor form an equidistant set of Caroli–de Gennes–Matricon (CdGM) energy levels with a tiny spacing $\omega_0 \sim \Delta/k_F \xi$ [1]. Since the product of the Fermi wave vector $k_F$ and coherence length

$\xi$ is typically very large for all usual superconductors, the level spacing $\omega_0$ turns out to be much smaller than the bulk gap $\Delta$, indicating that the spectrum can be considered as nearly continuous. Potential disorder reshuffles CdGM levels, while keeping their average density intact; the corresponding detailed solution was found long ago in Ref. [2]. The presence of quasiparticle excitations in the vortex core characterized by a finite DoS at the Fermi energy has important implications regarding dissipation in a mixed state of type-II superconductors at low temperatures. Motion of vortices driven by a *dc* transport supercurrent leads to parametric modulation of the CdGM states and their excitation from the lower to higher energy levels, with a subsequent inelastic relaxation [3]. Due to gapless nature of localized CdGM states, energy dissipation occurs at any small vortex velocity and thus leads to Ohmic conductivity in the flux-flow state [4]. The same mechanism is responsible for enhancement of microwave losses in the mixed state at low frequencies, $\omega \ll \Delta/\hbar$.

The above classical picture was questioned recently due to unexpected experimental results for *ac* dissipation at low vortex density in disordered Aluminum films [5]. It was found that while less disordered films followed the standard expectations, *ac* dissipation for the most disordered film was suppressed nearly by a factor of 40. It is difficult to explain this effect other than assuming the absence of the low-energy CdGM states in the most disordered Al film. It is not the first example of somewhat anomalous nature of electron states in a vortex core: The absence of CdGM states was reported by STM study in a copper-oxide high-temperature superconductor [6]. Besides that, numerical evidence for anomalous structure of energy levels of the vortex in a very strongly disordered superconductor was provided in Ref. [7]. No clear physical picture explaining these anomalies was proposed, as far as we are aware of. In the present paper we demonstrate one of possible mechanisms leading to absence of low-energy excitations inside a vortex core related to a granular nature of the superconducting material.

In a granular material, metallic grains are separated by tunnel barriers. The difference between diffusive and tunnel transport in mesoscopic electronic structures was discussed in details in Refs. [8]. It was shown, in particular, that the properties of a structure made of a piece of diffusive metal (resistance $R_D$) connected in series with a tunnel junction (resistance $R_T$) depends on the ratio of resistances, $R_T/R_D$. Perfectly conducting channels with transparencies $\mathcal{T}_n \to 1$ exist as long as $R_T < R_D$ and disappear in the opposite limit. On the other hand, it is exactly the presence of "arbitrary transparent" channels in a diffusive metal described by Dorokhov's distribution $P(\mathcal{T})$ [9] that leads to the absence of a minigap in an SNS junction with the phase difference $\varphi = \pi$ between superconducting terminals. The last statement can be illustrated by Beenakker's formula [10] for energy levels inside a short SNS junction: $E_n = \Delta[1 - \mathcal{T}_n \sin^2(\varphi/2)]^{1/2}$. A junction of an SINS type with a highly resistive tunnel barrier does not support conducting channels with $\mathcal{T}_n \approx 1$ and the gap in the excitation spectrum never closes.

The physics of SNS junctions at phase difference $\pi$ is similar in many respects to the physics of vortices, since electrons in both systems *on average* feel a zero order parameter [11]. It is therefore natural to expect that the quasiparticle spectrum inside a vortex in a superconductor composed of *small* grains connected by tunnel junctions between them may be qualitatively different from the CdGM gapless spectrum. An analogous effect is well known in the case of *large* superconducting islands with a well-defined intrinsic superconductivity coupled by weak Josephson junctions. Magnetic field penetrates such a structure in the form of core-less Josephson vortices localized near the junctions, these vortices do not host any low-energy excitations.

The situation with granular Al consisting of very small grains with the size of $l \approx 3 - 4$ nm [12] is somewhat intermediate. These grains are too small for superconductivity to exist in an isolated grain [13], since the corresponding level spacing $\delta = (2\nu_0 l^3)^{-1}$ exceeds the bulk superconducting gap $\Delta$, see Sec. 5 for the estimates ($\nu_0$ is the single-electron DoS at the

Fermi level per one spin). Hence it is inter-grain tunneling transport that is responsible for establishing superconductivity in granular Al. Therefore tunneling coupling cannot be weak that excludes possibility to describe material properties in terms of a perturbation theory in the tunneling Hamiltonian.

We will model polycrystalline media by a periodic set of metallic grains coupled through identical tunnel barriers. Electron dynamics in each grain will be assumed chaotic, either due to impurity scattering in the grain or due to random scattering on the boundaries (the latter case is probably realized for granular Al). The effective Thouless energy of a grain, $E_{Th} = \hbar/\tau_{erg}$, determined by the time $\tau_{erg}$ needed to travel across the grain is assumed to be much larger than the inelastic level width $\gamma = \hbar/\tau_{dwell}$ due to tunneling to a neighboring grain. Under this condition one may neglect spatial variations of the electron Green functions inside each grain and describe them in the zero-mode approximation [14]. Note that it is the inter-grain tunneling rate $\gamma$ that determines the macroscopic diffusion coefficient $D \sim \gamma l^2$, regardless of the details of the intra-grain electron dynamics. For this reason, the effective coherence length $\xi(\gamma)$ is a function of $\gamma$. Center of vortex is always located at the corner between three neighbouring grains, to minimize vortex energy.

In such a model, we will derive a discrete version of the Usadel equations [15] for electron Green functions in the superconducting state, and solve them in the presence of a vortex. The key parameter of our theory is the ratio of the effective coherence length $\xi(\gamma)$ to the distance between centers of neighboring grains $l$. For $\xi/l \gg 1$, discreteness of the problem is irrelevant and the quasiparticle spectrum does not differ from the one found in Ref. [2] for a usual disordered superconductor. With decreasing the transparency $\gamma$, the coherence length $\xi$ decreases and at $\xi/l < \zeta_c$ we find a gap in the excitation spectrum, with its magnitude growing with $\xi/l$ decrease. The critical value $\zeta_c$ is non-universal; numerically we found $\zeta_c \approx 1.4$ for the model of triangular array of hexagonal grains.

The rest of the paper is composed as follows: in Sec. 2 we derive the discrete Usadel and self-consistency equations for a 2D model of a granular superconductor. The spacial distribution of the order parameter in presence of a vortex is calculated in Sec. 3. Section 4 is devoted to the computation of the spatially resolved and integral density of states for various values of our key parameter $\xi/l$. In Sec. 5 we establish the conditions on the film resistance needed to have a gapless core. Finally, Sec. 6 contains our conclusions.

## 2 Discrete Usadel and self-consistency equations

We start from the action for granular superconducting system in the Matsubara formalism assuming Green functions $\hat{Q}_i$ to be uniform within each $i^{th}$ grain:

$$S[Q] = \frac{\pi}{\delta}\left[-\sum_i \text{Tr}(\varepsilon\hat{\tau}_3 + \hat{\Delta}_i)\hat{Q}_i - \gamma\sum_{\langle ij \rangle}\text{Tr}\hat{Q}_i\hat{Q}_j + \sum_i \frac{|\Delta_i|^2}{\pi\lambda T}\right], \qquad (1)$$

where $\Delta_i = |\Delta_i|e^{i\varphi_i}$ is the order parameter in the grain number $i$, $\hat{\Delta} = \tau_+\Delta + \tau_-\Delta^*$, parameter $\gamma$ measures tunneling conductance between grains, $\delta$ is the mean level spacing inside grain, $T$ is the temperature, and $\lambda$ is the dimensionless Cooper coupling constant. In the zero-mode approximation, when the intra-grain electron dynamics is irrelevant, the $Q$-part of the action in each grain acquires a universal random-matrix form [14]. Summation in the second term of the action (1) goes over all nearest-neighbouring pairs of grains connected by tunnel junctions. While in real granular metal grain's geometry and location are random, we will employ the simplest 2D model where each grain is a hexagon of fixed size and their centers are packed into the triangular lattice with the lattice constant $l$, see Fig. 1.

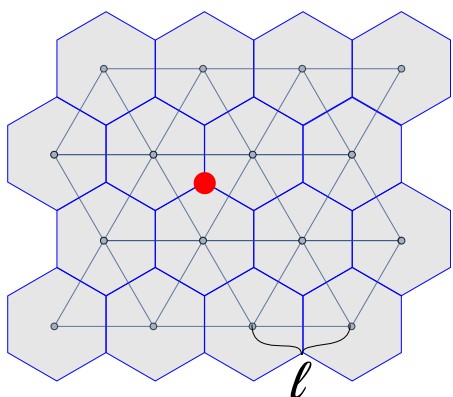

Figure 1: Sketch of the considered model. Red dot designates the center of the vortex that lies on the edge of three neighbouring grains as such location minimizes the free energy.

Although real granular arrays are more 3D-like usually, our 2D model makes sense since any nontrivial spatial dependence develops in the 2D plane transverse to the applied magnetic field. Still, there are some differences between bulk and 2D situations, and we will comment on this issue in the Discussion part of the paper.

Green functions in the grains are normalized as $\hat{Q}^2 = \hat{1}$, therefore the tunneling term written as $-\gamma\,\mathrm{Tr}\,\hat{Q}_i\hat{Q}_j$ is equivalent to $(\gamma/2)\,\mathrm{Tr}(\hat{Q}_i-\hat{Q}_j)^2$, which is a discrete version of the usual gradient term $D\,\mathrm{Tr}(\nabla\hat{Q})^2$, with the effective diffusion coefficient $D = 3\gamma l^2$ for the triangular lattice shown in Fig. 1. The saddle-point equations corresponding to the action (1) read as $[\hat{Q}_i, \delta S/\delta\hat{Q}_i] = 0$, due to the constraint $\hat{Q}^2 = \hat{1}$. The saddle-point solution $\hat{Q}_i$ is diagonal in Matsubara energies. In the angular representation it is given by

$$\hat{Q}_i = \begin{pmatrix} \cos\theta_i & e^{i\chi_i}\sin\theta_i \\ e^{-i\chi_i}\sin\theta_i & -\cos\theta_i \end{pmatrix},$$

where the spectral angle $\theta_i(\epsilon)$ and phase $\varphi_i(\epsilon)$ are energy-dependent. Then the saddle-point equations acquire the form of the discrete Usadel equations:

$$\begin{cases} \gamma \sum_{j:\langle ij\rangle} \sin\theta_j \sin(\chi_i-\chi_j) = -\sin(\chi_i-\varphi_i)|\Delta_i|\,, & (2) \\[2mm] \gamma \sum_{j:\langle ij\rangle} \left[\cos(\chi_j-\chi_i)\cos\theta_i\sin\theta_j - \sin\theta_i\cos\theta_j\right] - \epsilon\sin\theta_i + \cos\theta_i|\Delta_i|\cos(\varphi_i-\chi_i) = 0\,. & (3) \end{cases}$$

Varying the action over $\Delta^*$, one supplements the Usadel equations with the self-consistency equation for the order parameter:

$$\Delta_i = \pi\lambda T \sum_{\epsilon=-\omega_{\mathrm{D}}}^{\omega_{\mathrm{D}}} e^{i\chi_i}\sin\theta_i\,, \tag{4}$$

where $\omega_{\mathrm{D}}$ is the Debye energy. Equation (4) should be solved with the boundary conditions corresponding to the phase singularity located at the corner intersection of three grains, as shown in Fig. 1; this point have coordinates $(0,0)$.

Equations (2), (3), and (4) constitute the set of self-consistence equations which describe granular superconductor in the saddle-point approximation (in other terms, with dynamic fluctuations of $Q$-matrices being neglected). We provide quantitative criterion for validity of

this saddle-point approximation in the Discussion part of the paper; right now we just note that this approximation is valid as long as inter-grain coupling energy $\gamma$ is not too weak.

In the most general case, phases of the order parameter $\varphi_i$ and of the Green function $\chi_i(\epsilon)$ (in the same grain) might be different. However, for tunnel junctions between grains (the case we consider here) the difference between these phases is very small, as it contains higher powers of transmission coefficients (which are all small in the tunnel junctions), see Ref. [16] for a detailed discussion of this issue. Therefore we safely neglect that small difference and set $\chi_i(\epsilon) \equiv \phi_i$. The phase $\varphi_i$ of the $i^{\text{th}}$ grain is given by its vortex solution:

$$\varphi_i = \arctan(y_i/x_i). \tag{5}$$

Here vector $\mathbf{r}_i = (x_i, y_i)$ goes from the singularity point $(0,0)$ to the center of the $i^{th}$ grain. Hence we only have to check that equation (2) is satisfied after solving the simplified version of equations (3) and (4)

$$
\begin{cases}
|\Delta_i| = \pi \lambda T \displaystyle\sum_{\epsilon_n = -\omega_{\text{D}}}^{\omega_{\text{D}}} \sin \theta_i \,, & (6) \\[2ex]
\gamma \displaystyle\sum_{j:\langle ij \rangle} \left[ \cos\left(\varphi_j - \varphi_i\right) \cos\theta_i \sin\theta_j - \sin\theta_i \cos\theta_j \right] - \epsilon \sin\theta_i + \cos\theta_i |\Delta_i| = 0 \,. & (7)
\end{cases}
$$

Now we have to check that solution for $\theta$ satisfies (with good numerical precision) Eq.(2). The coupling constant $\lambda$ is related with the Debye energy $\omega_{\text{D}}$ via the BCS relation $\lambda = \ln(1.14\,\omega_{\text{D}}/T_c)$, where $T_c$ is the critical temperature. Our goal now is to find a vortex-like solution for the order parameter distribution, and then we should solve Usadel equation (7) at real energies $E$ (that is, after replacement $\epsilon_n \to iE$), to find DoS normalized over density of states in a normal state as $\nu_i(E)/\nu_0 = \operatorname{Re} \cos \theta_i(E)$.

## 3 Vortex solution for the order parameter

We solve numerically the system of equations (6) and (7) iteratively, using the following axially-symmetric Ansatz for the order parameter distribution and the spectral angle:

$$\Delta_i = \Delta_0 \tanh \frac{r_i}{\xi}, \qquad \theta_i(\epsilon) = \arctan\left(\frac{\Delta_0}{\epsilon} \tanh \frac{r_i}{\xi_\epsilon}\right). \tag{8}$$

The Ansatz for $\Delta_i$ is chosen to interpolate between the linear behavior at $r \to 0$ and uniform asymptotics at infinity, $|\Delta(r \to \infty)| \to \Delta_0$ where $\Delta_0$ - is value of order parameter in bulk continuous case without vortex. The parameter $\xi$ plays the role of the coherence length and will be optimized by the iterative procedure. The Ansatz for $\theta_i(\epsilon)$ is chosen in a similar way, it contains a set of energy-dependent lengths $\xi(\epsilon_n)$ to be optimized as well.

For the purpose of numerical study it is more convenient, instead of direct solution of Eqs. (6) and (7), to minimize the action (1) over the Anzatz parameters $\xi, \xi_\epsilon, \theta_i(\epsilon)$ after substitution of the matrices $\hat{Q}_i$ and the order parameter $\Delta_i$ in the form (8) into the action. We perform this procedure for many different values of the interface transparency $\gamma$ at low temperatures $T \ll T_c$ (numerical computation was performed for $T = 0.1\,\text{K}$ assuming $T_c = 2.2\,\text{K}$ for bulk Al). The obtained dependence of $\xi(\gamma)$ is shown in Fig. 2. Since the usual expression for the coherence length in the dirty limit is $\xi = \sqrt{D/2\Delta_0}$, and in our problem $D \propto \gamma$, we expect $\xi(\gamma) \propto \gamma^{1/2}$ at large $\xi/l \gg 1$, which agrees with the numerical result shown in Fig. 2.

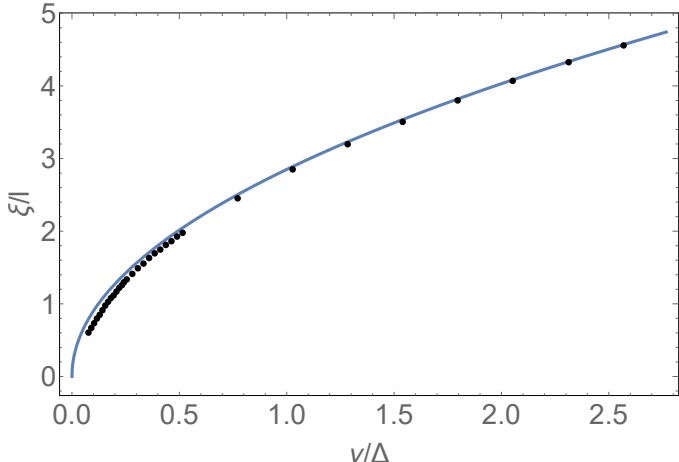

Figure 2: Dependence of the effective coherence length $\xi$ on the tunneling parameter $\gamma$ that determines the macroscopic diffusion constant $D$. Solid line is a guide for the eyes that represents the square root dependence for $\gamma \gg \Delta$.

## 4 Density of states: Minigap opening

With the obtained solution for $\Delta(r)$, we can now find the spatially resolved DoS by solving the Usadel equation (7) at real energies, i.e. after replacement $\epsilon_n \to iE$. Anticipating the minigap opening, it is useful to change the variable as $\theta = \pi/2 + i\psi$ [17]. Then in terms of the new variable $\psi_i$ the discrete Usadel equation (7) reads

$$\gamma \sum_{j:\langle ij\rangle} \left[\cos\left(\varphi_i - \varphi_j\right)\sinh\psi_i \cosh\psi_j - \cosh\psi_i \sinh\psi_j\right] - E\cosh\psi_i + |\Delta|\sinh\psi_i = 0, \quad (9)$$

where $\varphi_i$ is given by Eq. (5). Below we present our results for the space-resolved DoS $\nu_i(E)/\nu_0 = \operatorname{Im} \sinh\psi_i(E)$.

We start with Fig. 3, where we show typical dependencies of the DoS $\nu(E, r_i) = \nu_i(E)$ on the distance $r_i$ from the center of an $i$th grain to the vortex center calculated at several energies $E$. The results are provided for two quite different choices of the coherence lengths: (a) $\xi/l = 4.6$ ($\gamma/\Delta_0 = 2.2$), and (b) $\xi/l = 0.63$ ($\gamma/\Delta_0 = 0.08$). When the grains are well coupled and $\xi \gg l$, we obtain gapless $\nu(r, E)$ distributions similar to those found for a uniformly disordered superconductor [2], see Fig. 3(a). With decreasing the transparency of the intergrain boundaries and decreasing $\xi$, at $\xi \sim l$, we see a qualitatively different behavior, with the gap opening at low energies $E < E_g$, see Fig. 3(b). Namely, the DoS is identically zero at $E < 0.33\,\Delta_0$, while it is finite, yet small, at $E = 0.34\,\Delta_0$ (see a peak at $r/\xi \sim 1$). In other terms, the solution we obtain points out to the development of a minigap $E_g \approx (0.34\pm0.01)\Delta_0$ in the spectrum of localized excitation in the vortex core, for the parameters of Fig. 3(b).

A different way to visualize the minigap is presented in Fig. 4, where we plot the DoS as a function of energy at different distances from the vortex center, for the same two values of $\xi/l$ as in Fig. 3. When there are many grains in the core, at $\xi/l \gg 1$, the spectrum is gapless, with a finite DoS down to the Fermi energy, $E = 0$, see Fig. 4(a). At $\xi/l \approx 0.6$, the effects of granularity becomes important leading to a vanishing DoS below $E_g \approx 0.33\Delta_0$ for all distances $r$, demonstrating the absence of low-energy excitations.

Low-temperature dissipation during vortex motion is determined by the low-energy global DoS associated with a vortex. In Fig. 5(a) we plot the integral DoS

$$\nu_I(E) = \sum_i \nu(E, r_i) \quad (10)$$

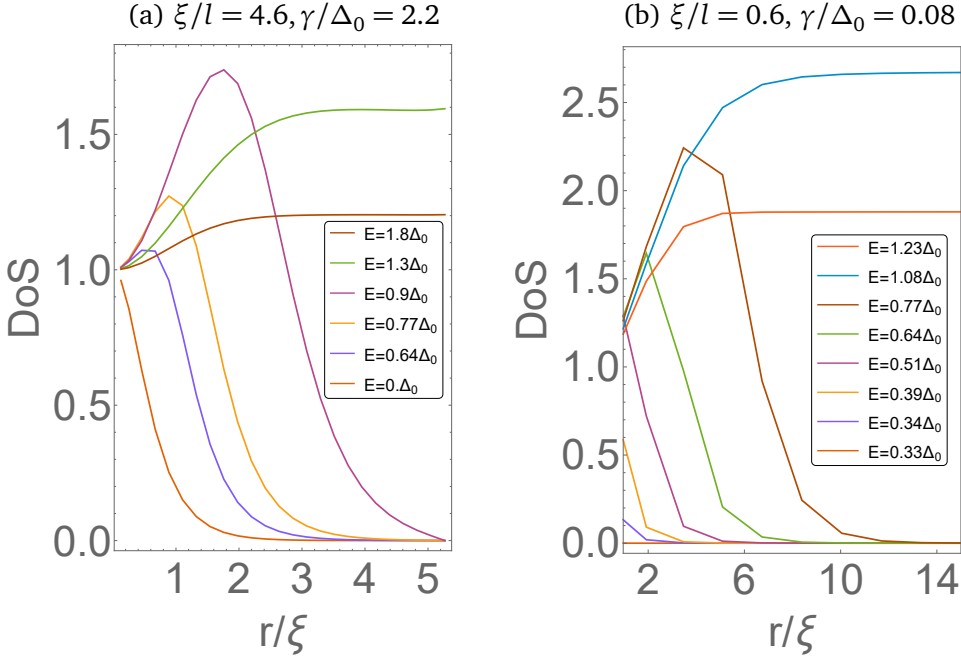

Figure 3: Normalized local DoS $\nu(E,r)/\nu_0$ as a function of the distance to the center of a vortex, $r$, at different energies for (a) $\xi/l \approx 4.6$ and (b) $\xi/l = 0.6$. With many grains in the core, $\xi/a \gg 1$, the solution is similar to that in the continuous limit found in Ref. [2]. At $\xi/l \sim 1$, there appears a minigap $E_g$ with a zero DOS at $E < E_g$. A piece-wise shape of the curves is due to discretness of the values of $r_i$ for the lattice.

in the subgap region ($E < \Delta_0$) normalized by $\nu_0 n_{core}$, where $n_{core} = \pi\xi^2/S_{gr}$ is the number of grains within the core, and $S_{gr} = \sqrt{3}l^2/2$ is the grain area. For large enough $\xi/l > 1.5$ the curves nearly overlap, following the behavior known for continuous Abrikosov vortices [2]. At smaller values of $\xi/l$ a minigap in the spectrum is clearly visible, with its magnitude growing with the decrease of $\xi/l$. Figure 5(b) demonstrates the same quantity $\nu_I(E)$ at $E > \Delta_0$, which is now normalized by the whole area of the system, since the corresponding eigenstates are delocalized. Here we see that starting from $E/\Delta_0 \geq 1.2$ the effects of granular structure are nearly invisible, while at lower energies $\nu_I(E)$ is enhanced at small $\xi/l$.

Analysing the data like those shown in Fig. 5(a) for a number of different values $\xi/l$, we found that the minigap opens at $\xi/l = \zeta_c$, with $\zeta_c \approx 1.4$. The dependence of the minigap magnitude $E_g$ on $\xi/l$ for $\xi/l < \zeta_c$ is shown in Fig. 6(a). In addition, in Fig. 6(b) we demonstrate evolution of the integral DoS $\nu_I(0)/\nu_0$ at zero energy as a function of $\xi/l$ in the range $\xi/l > \zeta_c$. Note that $\nu_I(0)$ grows very sharply in the narrow range of $\xi/l \gtrsim \zeta_c$ just above the threshold value.

## 5 Discussion

In this section we obtain an estimate for the range of parameters where gapful vortices could be observed. While the level width $\gamma$ is the main parameter, which controls the tunnel coupling between neighboring grains, for practical purposes it is more convenient to work in terms of the normal-state sheet resistance of the film, $R_\square$. The latter is related to $\gamma$ via Einstein's relation for the bulk normal-state conductivity $\sigma = 2\nu_0 e^2 D$ (here the factor 2 accounts for the electron spin) and the expression for diffusion coefficient $D = 3\gamma l^2$. Hence we obtain $e^2 R_\square/\hbar = (6\nu_0\gamma l^2 d)^{-1}$, where $d$ is the film thickness.

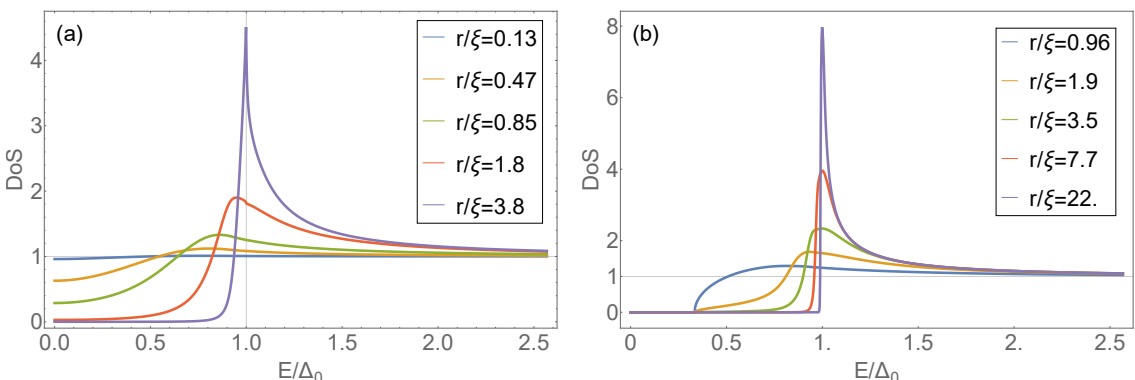

Figure 4: Normalized local DoS $\nu(E, r)/\nu_0$ as a function of energy $E$ for grains at different distances to the vortex center, for (a) $\xi/l \approx 4.6$ and (b) $\xi/l \approx 0.6$, as in Fig. 3. At $\xi/l \gg 1$ and spectrum is gapless, while at $\xi/l \lesssim 1$ there is a minigap with vanishing DoS for $E < E_g$.

In what follows we will consider a film as consisting of $d/l$ layers of grains of thickness $l$ each, so that $d \geq l$. The level spacing inside each grain is then $\delta = (2\nu_0 S_{\mathrm{gr}} l)^{-1}$. Now writing the coherence length as $\xi^2 = \hbar D/2\Delta_0$, we can represent the condition $\xi/l < \zeta_c$ for the existence of a minigap in terms of $R_\square$ and the ratio $\delta/\Delta_0$ in the form

$$\frac{e^2 R_\square}{\hbar} > 0.2 \frac{l}{d} \frac{\delta}{\Delta_0}, \tag{11}$$

where we used $\zeta_c \approx 1.4$ numerically derived for Al. Using the normal-state DoS $2\nu_0 = 2.15 \times 10^{34}\,\mathrm{erg}^{-1}\mathrm{cm}^{-3}$, we estimate the level spacing for Al grains with the size $l = 4\,\mathrm{nm}$ as $\delta \approx 8.4 \cdot 10^{-16}\,\mathrm{erg}$, which is 3 times larger than the gap $\Delta_0 \approx 3.9 \cdot 10^{-16}\,\mathrm{erg}$.

Another condition comes from the requirement already mentioned in Introduction that the tunnelling coupling $\gamma$ should not be too low, otherwise mean-field description of superconductivity will not be adequate and superconducting pairing will be suppressed due to level quantization within single grain. The condition that allows to use usual approach with a self-consistent order parameter can be found by comparing level spacing $\delta$ and coupling energy between a grain and its surrounding, equal to $6\gamma$ for the triangular array shown in Fig. 1 where

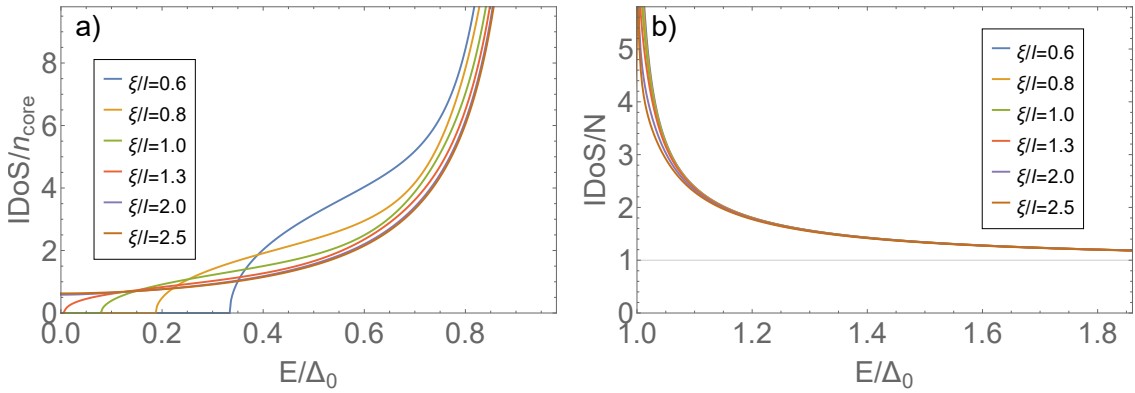

Figure 5: Integral DoS $\nu_I(E)/\nu_0$ as a function of energy at different values of the granularity parameter $\xi/l$: (a) subgap energies $E < \Delta_0$, data normalized by $\nu_0 n_{\mathrm{core}}$; (b) higher energies $E > \Delta_0$, data normalized by $\nu_0$ times the total number of grains in the system.

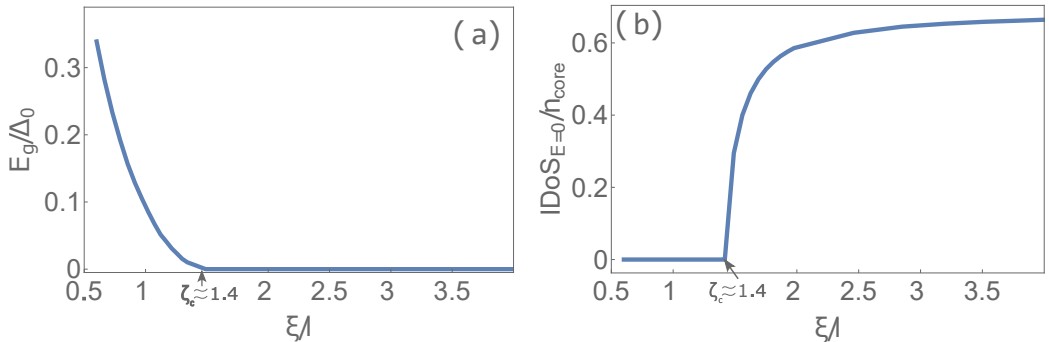

Figure 6: (a) The minigap $E_g$ as a function of $\xi/l$, vanishing at $\xi/l = \zeta_c \sim 1.4$. (b) Dependence of the (normalized) integral DoS at zero energy on $\xi/l$.

each grain has 6 nearest neighbors. On a more formal level, the argument is as follows: to be able to treat the action (1) within saddle-point approximation, we need to have the action cost for $\hat{Q}$-matrix fluctuations to be large, which means $\delta$ should be smaller than either $\Delta$ (which is not the case here), or $6\gamma$. In terms of the sheet resistance, the resulting condition $\delta < 6\gamma$ reads, for purely 2D array with $l = d$, as

$$\frac{e^2 R_\square}{\hbar} < 1 \,, \tag{12}$$

which is compatible with Eq. (11) for $l = d$ case if $\delta$ does not exceed $4\Delta_0$. Hence a gapful vortex state is expected to exist in a resistivity window

$$0.2 \frac{\delta}{\Delta_0} < \frac{e^2 R_\square}{\hbar} < 1 \,, \tag{13}$$

for pure 2D granular film. For thicker films with $d \gg l$ the right inequality in Eq. (13) should be modified since the number of nearest neighbours in a typical dense 3D arrays is about 10-12 instead of 6. In result, the 3D analog of Eq. (13) reads as

$$0.2 \frac{\delta}{\Delta_0} \frac{l}{d} < \frac{e^2 R_\square}{\hbar} \le \frac{2l}{d} \,, \tag{14}$$

which makes the range of applicability of our approach broader for thicker films in comparison to pure 2D ones.

A separate issue to be discussed is related with intrinsic inhomogeneity of natural granular films. First of all, let us discuss available experimental results concerning location of the superconductor-insulator transition in granular Al. The data provided in Ref. [18] show that superconducting state survive in relatively thick films with resistivity up to $\rho \approx 10^4 \, \mu\Omega$ cm, while the film "H" with $\rho = 3000 \, \mu\Omega$ cm and thickness 30 nm is located relatively far inside superconducting domain (see Fig. 3 of Ref. [18]), with $T_c \approx 2$K. Dimensionless conductance of this film is $e^2 R_\square/\hbar \approx 0.25$ while $2l/d \approx 0.27$ in the R.H.S. of Eq. (14). This comparison tells us that in reality the condition for well-developed superconductivity to exist (and to be described by self-consistent approach) is less stringent than Eq.(14) indicates. Now it is worth to discuss the role of inhomogeneity of granular films in formation of a spectral gap. The major kind of inhomogeneity is provided by spatial fluctuations the coupling strengths $\gamma_{ij}$. In result, stronger junctions will form larger clusters of initial small grains, while weaker couplings will form junctions between those clusters. Macroscopic conductivity of the film is controlled by the *typical* coupling strength $\gamma_{\rm typ}$. Fluctuations in actual values of $\gamma_{ij}$ lead therefore to the increases of effective size of clusters (playing now the role of effective grans) which enter into

our theory. In means, in turn, that the left inequality in Eq. (14) will be replaced by somewhat less stringent condition. To summarize: spatial disorder of real granular media makes wider the parameter region where gapful vortices can be found.

We checked also that our main result for the critical value $\zeta_c$ is left unchanged with respect to slight variation of the BCS coupling constant $\lambda$ between the values corresponding to transition temperature of clean Al($T_c = 1.2$K) and granular one ($T_c = 2.2$K).

# 6 Conclusions

We demonstrate that gapless electron states are absent inside the vortex core in a granular superconductor with moderately weak coupling between grains, contrary to their classical counterparts [1, 2]. The magnitude of the minigap is computed for a specific model of a granular superconductor with triangular lattice of identical grains. A very similar effect of gap opening in the spectrum of quasiparticle states localized in the vortex core has been recently reported for a different problem of a vortex in a clean superconductor in the presence of a planar defect [19, 20]. This points out a crucial role of extended defects in breaking the continuity of the chiral branch of low-energy states in the vortex core.

In terms of the normal-state resistance of the film, for vortices without low-energy excitations to exist, two conditions should be satisfied, as given by Eq. (13). For the magnitude of the microwave quality factor $Q(\omega)$, the key issue is the comparison between $\hbar\omega$ and the minigap $E_g$, those behavior is shown in Fig. 6(a) as a function of $\xi/l$ for our model of identical grains connected by identical junctions. In a real granular Al couplings $\gamma_{ij}$ between grains fluctuates; relatively strongly coupled grains may compose "supergrains" of larger size, which are themselves coupled together by weaker couplings. Qualitatively, such situation is even more favorable for the existence of "gapful vortices", so they can exist in a broader range of film's resistances.

An additional effect that may contribute to the observed [5] suppression of microwave losses is the increase of vortex pinning strength due to granularity; however, we do not expect this effect itself to be strong enough, as in the experiment $Q$-factor jumps up by the factor $\sim 100$.

Finally, we emphasize that in our idealized model the zero-energy DoS grows very sharp when the parameter $\xi/l$ exceeds its critical value, see Fig. 6(b). The same feature should be expected for the vortex-related dissipation as well. In a real granular metal, we expect this jump in dissipation to be smeared due to broad distribution of couplings $\gamma$.

# Acknowledgments

We acknowledge discussions of experimental results with B. L. T. Plourde and K. Dodge, and theoretical model with A. S. Osin.

**Funding information**   This research was supported by the Basis Foundation under Grant No. 21-1-1-38-4 (D.E.K. and M.V.F.) and by the Russian Science Foundation under Grant No. 20-12-00361 (M.A.S.).

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
