# Peer review of "Gapful electrons in a vortex core in granular superconductors"

_SciPost Physics, doi:SciPost Phys. 15, 008 (2023)_

## Round 1 · Referee Report · Anonymous · 2023-1-16

Report
This is important and timely contribution to the physics of vortex matter in superconductors. The manuscript presents the results of theoretical study of quasiparticle density of states (DoS) inside the vortex core in granular superconductors. The results generalize the classical solution applicable for dirty superconductors. By solving a discrete version of the Usadel equations the authors show that, in contrast to the clasasival soluition, the electron DoS exhibits an energy gap in the case when the vortex core size is comparable to the distance between neighboring grains . The results provide a mechanism of suppression of microwave dissipation in a mixed state of granular Al.
The manuscript is clearly written and will be of interest to a broad community of researches in the field of superconductivity.
I suggest the manuscript for publication in its present form.
Author: Mikhail Feigel'man on 2023-02-07 [id 3328]
(in reply to Report 3 on 2023-02-06)We agree with the comments made by the Referee and we will make requested small changes in the resubmitted manuscript.

---

## Round 1 · Referee Report · Anonymous · 2023-1-23

Strengths
1. Fine model ( though introduced without proper references)
2. Plasuible and interesting answer
3. A numerical approach is adeqate.
4. The model permits an adequate numerical approach
Weaknesses
1. An ad-hoc assumption upon the self-consistent order parameter.
2. An ad-hoc assumtion upon the Green function phases.
3. With these assumptions, the answer has all chances to be numerically incorrect and even misleading.
4. The applicability of the model to mutli-layer film is questionable.
5. The discussion in Ch. 5 is rather confusing on my taste.
Report
Generally, it is an interesting formulation of a research project that, regretably, has not been accomplished yet. It is my impression that the authors did not have either time or computer resource to provide an extensive numerical analysis beyond the ad-hoc assumptions they put forward. I sincerely wish they will find both and provide a better version of the manuscript.
Requested changes
Obviously, the authors should do all the calculations without imposing the ad-hoc assumptions mentioned. Technically, this is straightforward, though may be time-consuming. Let me list the features of the assumptions that I find especially dangerous with respect to obtaining correct and sensible answer. a. Delta under assumptions always dissapears at one of the granula. Does not have to be the case: the vertex core between the granulas is obviously energetically more favourable. b. Periodic arrangement. How sensitive is the answer to the lattice type/random arrangement c. The "core" of superconducting phases does not have to be in the same position as the "core" of \chi phases. There may be an energy gain associated with such separation.
Author: Mikhail Feigel'man on 2023-01-24 [id 3267]
(in reply to Report 2 on 2023-01-23)
Response to the part "Weaknesses" 1. An ad-hoc assumption upon the self-consistent order parameter.
Self-consistent order parameter is the basic theory concept in low-temperature superconductivity. It can be questioned only in very special cases like proximity to some quantum phase transition. Nothing like that is considered in our paper by construction of the model and due to physical content considered
An ad-hoc assumtion upon the Green function phases.
There are no "ad-hoc assumptions" here. The fact that phases of the Green function and of the order parameter coincide is proven for the systems with tunnel junctions; in particular detailed way it is done in the paper we cite as Ref.[16].
With these assumptions, the answer has all chances to be numerically incorrect and even misleading.
As shown above this statement has no ground.
The applicability of the model to mutli-layer film is questionable.
We consider this situation of magnetic field transverse to the film plane, thus the order parameter variations in the field direction are absent. That makes possible to use 2D model we employed for thick films as well.
The discussion in Ch. 5 is rather confusing on my taste.
This statement reflects personal feeling of its author but does not give us any possibility to understand what is wrong or incorrect or misleading in the text; therefore such a comment is not useful for scientific discussion.
Response to the part Requested changes
First statement of this part is based on the statements from the part "weaknesses" which are incorrect, as shown above. Next we comment to specific statements listed as
a) Delta under assumptions always dissapears at one of the granula
Incorrect statement concerning the content of our paper; vortex center is located between the grains, of course.
We will make it more evident in resubmitted version
b) Periodic arrangement. How sensitive is the answer to the lattice type/random arrangement
The answer is twofold: 1) for weak disorder in grain sizes or positions small change in the position of spectrum edge we found is expected; it will be weak since the gap we found is large, it is some valuable fraction of the full superconducting gap; 2) for strong disorder the problem should be considered separately, and this will be the subject of separate publication.
c) The "core" of superconducting phases does not have to be in the same position as the "core" of \chi phases.
This incorrect statement was commented above regarding the part2 of "weaknesses". These phases coincide due to tunnel nature of junctions between grains, see Ref.[16].

---

## Round 1 · Referee Report · Anonymous · 2023-2-6

Strengths
1. Clear formulation of the model
2. Reliable methods for the evaluation of the density of states within the formulated model
Weaknesses
1. The use of the term vortex size in the Abstract, together with notation \xi for it may confuse a potential reader. Perhaps it would be better to use superconducting coherence length in the granular medium as a parameter (\xi).
2. Given the limitations on the model parameters set forth by Eq. (12), it is better to tone down the claim about explaining low dissipation in granular Al films. In addition to modifying the density of states, granularity may affect the vortex motion casually mentioned in the manuscript.
Report
This is a publishable manuscript. Addressing the above concerns would not hurt.
Requested changes
Please see the Weaknesses

---

## Round 1 · Referee Report · Anonymous · 2023-4-3

(Invited Report)- Cite as: Anonymous, Report on arXiv:2212.01862v1, delivered 2023-04-03, doi: 10.21468/SciPost.Report.6991
Report
I have carefully read the manuscript and found that it contains the very interesting physical idea. It is aesthetically appealing and provides a new insight to the possible explanation of the recent experimental results on microwave spectra in granular superconducting films, thus I suggest this paper for a publication in SciPost and second to Referee 3.
I have also read previous referees' comments and authors replies and believe that majority of these comments are properly answered and when needed are incorporated into the revised 2nd version of the paper.
My minor suggestion (which is optional) is to add a small Appendix where authors may show the form of action, when the latter is expressed solely in terms of variational parameters (a coherence length, an energy dependent phase of the order parameter, etc.), because this action was in fact used to (approximately) solve the Usadel equations (2-3) together with the self-consistent equation (4) for the order parameter.

---

## Round 2 · Author Response

Dear Editor and Referees,
please find resubmitted version of our manuscript “Gapful electrons in a vortex core in granular superconductors”. The manuscript is considerably modified according to the reports of the referees 2 and 3 (referee 1 did not asked for any changes and supported publication in the original form). Detailed list of changes made is located in the end of the present file.
Below you find point-to-point answers to the points raised by the referees.
We hope the resubmitted version of the manuscript solves all the issues raised and will be accepted for publication. Please note that the subject of high-inductance and low-dissipation superconducting materials currently attracts high interest from many groups around the World. Please also note that two (out of all three) referees supported univocally the publication of our manuscript in SciPost – Physics.
Sincerely yours, D. Kiselov, M. Feigel’man and M. Skvortsov
Answers to the issues raised by the referees:
Referee 2 comments:
1. An ad-hoc assumption upon the self-consistent order parameter.
Self-consistent order parameter is the basic theory concept in low-temperature superconductivity. It can be questioned only in very special cases like proximity to some quantum phase transition. We added, to avoid any further misunderstanding, an extensive discussion concerning the conditions for the use of the mean-field (self-consistent) approach. This discussion is mainly located in the Section V, Discussions.
2. An ad-hoc assumtion upon the Green function phases.
There are no "ad-hoc assumptions" here. The fact that phases of the Green function and of the order parameter coincide is proven for the systems with tunnel junctions; in particular detailed way it is done in the paper we cite as Ref.[16]. We reformulated the corresponding piece of the text, to avoid any further misunderstanding.
3. With these assumptions, the answer has all chances to be numerically incorrect and even
misleading.
As shown above, this statement has no ground.
4. The applicability of the model to mutli-layer film is questionable.
We consider this situation of magnetic field transverse to the film plane, thus the order parameter variations in the field direction are absent. That makes possible to use 2D model we employed for thick films as well. We added detailed discussion of this issue into the Section II and Section V.
5. The discussion in Ch. 5 is rather confusing on my taste.
This statement reflects personal feeling of its author but does not give us any possibility to understand what is wrong or incorrect or misleading in the text; therefore such a comment is not useful for scientific discussion.
6. Delta under assumptions always dissapears at one of the granula
This is an explicit incorrect statement; vortex center is located between the grains, of course. We added additional text to explain this point once again; it is located in the end of the 2nd paragraph in the left column of page 2.
- Periodic arrangement. How sensitive is the answer to the lattice type/random arrangement
We added detailed discussion of this issue to the Sec. V. In short, randomness makes our results even more stable.
8. The "core" of superconducting phases does not have to be in the same position as the "core"
of \chi phases.
This incorrect statement was already commented above. The phases \chi_i(\epsilon) and \phi_i coinside due to the tunnel nature of junctions between grains, as we reiterated again in the paragraph of Sec.II that is situated above Eq.(5).
Referee 3 comments:
-
To replace “vortex size” by “coherence length” -
Done, in 2 locations.
-
To tone down the claim about explaining low dissipation in granular Al films -
Done, in the Abstract

---

## Round 2 · List of Changes

1. a. Abstract: stylistic changes aimed to explain actual outcome of the paper
more precisely were made.
b. The term “vortex size” was replaced by “coherence length”
2. First section: comment was added with the general explanation on the
position of the vortex in the end of the 2nd paragraph in the left column
on page 2.
3. Second section:
a. comment was included on usability of the model employed in real 3D material - in the end of the 1st paragraph in the right column of page2.
b. The paragraph (the one just before Eq.(5)) about approximations used for the phase variables was rewritten. Also, some explanation was added in the previous paragraph about validity of the saddle-point approximation.
4. Third section: minor rework was done to clarify the numerical approach
employed. Also, in the caption to Fig.2 the term “vortex size” was
replaced by “coherence length”
5. Section V: an additional discussion was provided about the role of disorder
in the inter-grain couplings. Major outcome of this qualitative consideration
is as follows: the region of validity of our main result becomes wider due to
inhomogeniety of parameters in real granular Al films. This conclusion is
supported by the analysis of experimental data from Ref.[18].
6. Section VI: comment on another effect that can contribute to the suppression
of the microwave losses was added.
7. Bibliography: A citation [18] of the recent experiment on optical response,
superconductivity and resistance in granular Al is added.
Anonymous on 2023-04-03 [id 3537]
this is a report of referee 4 on the resubmitted manuscript, entered into the fields for the previous version by mistake; it is copied here by the editor.
I have carefully read the manuscript and found that it contains the very interesting physical idea. It is aesthetically appealing and provides a new insight to the possible explanation of the recent experimental results on microwave spectra in granular superconducting films, thus I suggest this paper for a publication in SciPost and second to Referee 3.
I have also read previous referees' comments and authors replies and believe that majority of these comments are properly answered and when needed are incorporated into the revised 2nd version of the paper.
My minor suggestion (which is optional) is to add a small Appendix where authors may show the form of action, when the latter is expressed solely in terms of variational parameters (a coherence length, an energy dependent phase of the order parameter, etc.), because this action was in fact used to (approximately) solve the Usadel equations (2-3) together with the self-consistent equation (4) for the order parameter.
Anonymous on 2023-02-24 [id 3404]
this is a report of referee 3 on the resubmitted manuscript; the report was sent by email, and is entered by the editor
I have read Report 2 doubting the validity of the manuscript. My impression is that some of the doubts are based on a misunderstanding. The Referee's apparent misconception is that the Authors consider a vortex with a core inside a grain. This is not the case: the Authors essentially generalize the notion of a Josephson junction vortex onto a granular medium. A vortex in their manuscript is a certain arrangement of the phase differences of the order parameter in neighboring grains. This unconventional structure of the core actually explains the main claim of that manuscript: under certain conditions, a vortex core in a granular medium does not lead to a gapless spectrum, unlike the vortices in continuous superconductors. I believe the main qualitative conclusions of the manuscript are correct.
In fairness to the Referee, the Authors could spend more effort on exploring whether their results are sensitive to the geometry of the granularity model. It is hard to expect though a strong sensitivity.

---

## Editorial Decision

published